# Beyond End-to-End Models: Characterizing the Favorable Scaling of Coordinated Perception and Control

## Abstract

End-to-end models have become a dominant paradigm for learning in embodied agents, directly mapping raw sensory inputs to control outputs. However, their tightly coupled nature often leads to unfavorable scaling properties: as model size and environmental complexity increase, their computational and sample efficiency drastically deteriorates, posing a barrier to sustainable, long-term deployments in the real world. In this work, we move beyond the end-to-end approach and systematically characterize the scaling laws of an alternative paradigm: a coordinated architecture that decouples perception and control. In this architecture, perception and decision-making are handled by distinct agents that interact through a closed-loop feedback mechanism, where task-level feedback from control agents continuously guides the online evolutive learning of the perception modules. We conduct the first empirical study that investigates the computational scaling of these two paradigms across two critical dimensions: increasing model parameter scale and escalating environmental complexity. Our experiments show that the coordinated perception and control architecture achieves nearly linear or sub-linear growth (scaling exponents $\alpha \approx 1.11$ for parameters and $\delta \approx 0.95$ for complexity), while end-to-end models exhibit super-linear or exponential growth ($\alpha \approx 1.27$, $\delta \approx 2.70$). These results demonstrate that coordinated perception and control offers a fundamentally more scalable and robust design, providing practical guidance for building next-generation embodied AI systems that are both adaptive and computationally efficient.

## 1 Introduction

In recent years, end-to-end learning has become a dominant paradigm for intelligent systems that map perceptual inputs to decision outputs. The central idea is to employ a unified training pipeline that directly transforms high-dimensional sensory data into downstream planning or control commands Chen et al. (2024). This paradigm has received considerable attention in embodied intelligence applications such as robotics and autonomous driving. Notably, the design of end-to-end architectures has itself evolved over time. Early approaches were primarily monolithic models that integrated perception and decision-making into a single structure Pomerleau (1988); Bojarski et al. (2016); Akkaya et al. (2019); Chitta et al. (2022); Wu et al. (2022). As task complexity increased, many modern systems, particularly in autonomous driving, have shifted to modular end-to-end designs, where perception, prediction, and planning are separated into distinct submodules, as shown in Figure 1(a). Despite this modularization, these systems typically remain end-to-end in training, using joint backpropagation to optimize the entire system and thereby avoiding the error accumulation found in traditional fully decoupled pipelines.

Despite their widespread adoption and success, end-to-end paradigms are increasingly confronted with a growing scaling crisis. On one hand, pursuing higher accuracy often requires deeper architectures and auxiliary tasks, which rapidly increase model size, thus increases computational complexity Jia et al. (2025); Shao et al. (2024). On the other hand, improving generalization and performance in real-world environment demands massive training datasets and diverse simulation scenarios Dosovitskiy et al. (2017); Caesar et al. (2020; 2021); Jia et al. (2024); Dauner et al. (2024); Kolve et al. (2017); Savva et al. (2019); Dasari et al. (2019). These two pressures jointly amplify the

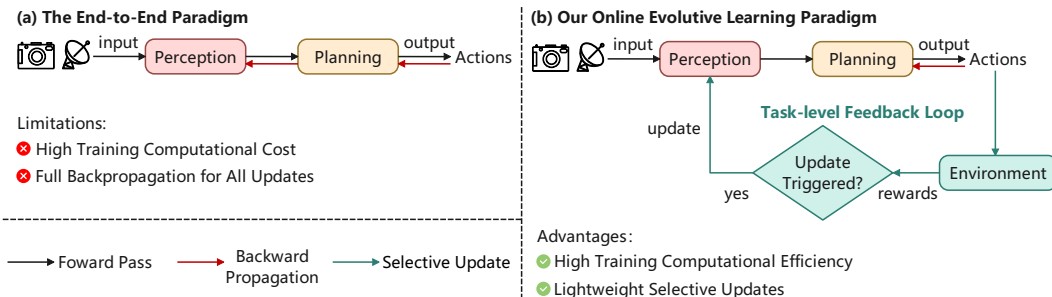

Figure 1: **Different training paradigms.** (a) The end-to-end paradigm's reliance on full back-propagation results in high computational costs during training. (b) Our online evolutive learning paradigm introduces a task-level feedback loop for lightweight selective updates, leading to a notable improvement in computational efficiency.

training cost of fully optimized pipelines, limiting iteration efficiency and deployment feasibility on resource-constrained platforms.

This challenge motivates a central question: *Can we design a framework that preserves the advantages of end-to-end learning while substantially improving training efficiency and reducing computational cost?* A straightforward answer would be to revert to fully decoupled pipelines. However, such methods are vulnerable to the classic error cascade problem, where small perception errors propagate downstream and undermine the robustness of planning Chen et al. (2024).

To address these challenges, and inspired by the idea of Online Evolutive Learning (OEL) Song et al. (2022), we propose a new framework called Coordinated Perception and Control framework. This approach strikes a balance between fully decoupled pipelines and strictly end-to-end optimization. While adopting a modular architecture, it introduces a closed-loop feedback mechanism that links perception and control during learning. Task-level feedback signals obtained from agent-environment interaction, such as rewards or costs, are used as high-level supervision to selectively and conditionally update perception modules. In practice, this means that targeted updates are triggered only when decision outcomes reveal perceptual deficiencies, rather than backpropagating through the entire system in every iteration. This mechanism retains adaptivity while substantially reducing the computational burden of training, as shown in Figure 1(b).

To systematically study the scaling behavior of different paradigms, we conduct experiments in a configurable grid-world environment. Although abstract compared to real-world settings, this environment allows precise control over variables such as model size and environmental complexity, enabling us to characterize computational scaling laws. We first formalize an evaluation metric for total training cost, and then benchmark our framework against end-to-end baselines across multiple model scales and environment complexities, in both single-agent and multi-agent settings. The results show that, compared with end-to-end approaches, our method achieves comparable or superior task performance while requiring far less training cost, with scaling exponents $\alpha \approx 1.11$ (OEL) versus $\alpha \approx 1.27$ (E2E) for model size, and $\delta \approx 0.95$ versus $\delta \approx 2.70$ for environment complexity. These findings highlight fundamental scaling properties and provide valuable insights for designing efficient architectures in more complex embodied intelligence domains.

Our contributions are as follows:

- We propose the coordinated perception and control framework, which augments modular architectures with a closed-loop feedback mechanism. Task-level feedback serves as high-level supervision to conditionally update perception modules, thereby retaining the advantages of end-to-end learning while greatly reducing training cost.

- We formalize an evaluation metric for total training cost and establish a methodology to analyze computational scaling laws. By leveraging a configurable grid-world environment, we systematically vary model scale and environment complexity, enabling precise characterization of scaling behaviors across different paradigms.

- We validate the proposed framework through extensive single-agent and multi-agent experiments. Results show that, compared to end-to-end baselines, our approach achieves comparable or superior task performance with more favorable scaling exponents ($\alpha \approx 1.11$ vs. $1.27$, $\delta \approx 0.95$ vs. $2.70$). These findings provide quantitative evidence for the efficiency and robustness of our paradigm, offering practical guidance for future algorithm design in complex scenarios.

## 2 RELATED WORKS

### 2.1 END-TO-END MODELS

The end-to-end learning paradigm, which directly maps raw inputs to final outputs, has become mainstream in modern AI. In computer vision, methods such as ResNet He et al. (2016) and Vision Transformer (ViT) Dosovitskiy et al. (2020) take images as input and output classifications. In NLP, transformer-based models like BERT Devlin et al. (2019) directly map question-context pairs to answers. In speech recognition, approaches such as Listen, Attend, and Spell (LAS) Chan et al. (2015) transcribe audio spectrograms into text sequences, unifying acoustic, pronunciation, and language models. In embodied intelligence, agents translate high-dimensional perceptual inputs into low-level control actions. For example, in autonomous driving, UniAD Hu et al. (2023) fuses multimodal sensor data to directly output future vehicle trajectories, while in robotic manipulation, models like Action Chunking with Transformers (ACT) Zhao et al. (2023) map multi-view images and proprioceptive states directly to action sequences. This paper focuses on end-to-end models for embodied agents that process high-dimensional perceptual inputs and output physical actions, trained via full network backpropagation. Despite their capabilities, such holistic training poses significant computational and data challenges.

Motivated by OEL in Song et al. (2022) and driving agents based on the OEL paradigm in Qian et al. (2024), we propose a coordinated perception and control training paradigm for embodied agents. Unlike the above works, we further conduct extensive experiments across diverse large-scale scenarios and compare our approach against end-to-end methods in terms of scaling characteristics.

### 2.2 SCALING OF END-TO-END PARADIGM

Recent work has begun to systematically study the scaling behavior of end-to-end models in reinforcement learning and application fields such as robotics and autonomous driving. Unlike large foundation models, these efforts focus on mid-sized embodied systems. On the model side, increasing capacity helps mitigate data inefficiency. Naumann et al. (2025) show that larger driving policies achieve the same accuracy with far less data, effectively shifting the scaling curve. On the data side, imitation learning studies show that policy performance improves sublinearly with dataset size, often following power-law trends. Lin et al. (2024) demonstrate that diversity of training environments and objects is far more critical than sheer demonstration count, with diminishing returns once basic coverage is achieved. Similar insights arise in multi-task robot learning Kalashnikov et al. (2021); Cai et al. (2025), where success depends on covering a wide range of conditions rather than accumulating redundant trials. However, these improvements demand substantially more computation, raising sustainability concerns. Despite these advances, major inefficiencies remain. For example, closed-loop studies in autonomous driving reveal steep scaling costs: even modest improvements in trajectory error require tens of thousands of additional training hours Naumann et al. (2025). Besides, in RL, high-parallel regimes expose gradient conflicts and replay inefficiencies, motivating techniques such as curriculum learning, conflict-aware gradient updates, and value-function regularization Joshi et al. (2025).

Prior work on scaling end-to-end systems mainly examines model size or dataset scale versus performance, with relatively little attention to how training cost grows when both model parameters and environment complexity increase. In contrast, our work (i) systematically analyzes computational scaling under these two factors, and (ii) introduces a coordinated perception and control framework that improves efficiency by updating perception modules only when control feedback indicates deficiencies.

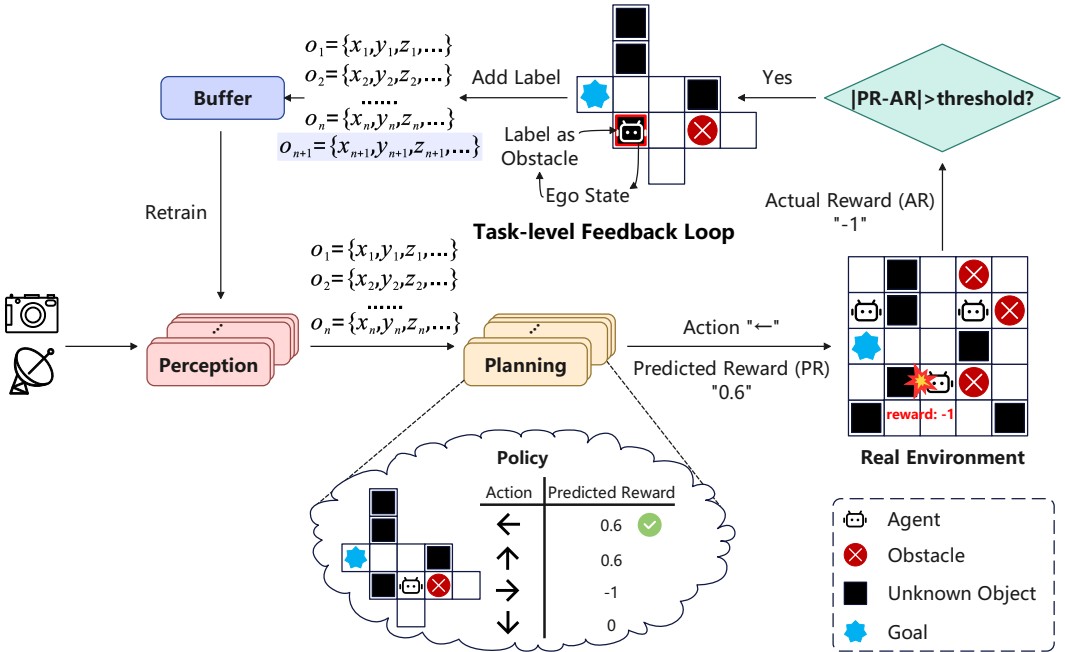

Figure 2: **Coordinated perception and control framework.** The process begins with the perception module interpreting sensory data and passing its results to the planning module. Based on its policy, the planning module then outputs a selected action along with a Predicted Reward (PR). This action is executed in the real environment, yielding an Actual Reward (AR). Then the PR and AR are compared, and if the discrepancy exceeds a predefined threshold, a selective update is triggered. This allows the system to use task-level outcomes to relabel the problematic sensory data and efficiently fine-tune the perception module online, thereby closing the feedback loop.

## 3 METHODS

Our proposed coordinated perception and control framework based on OEL is composed of three key components: a perception module that interprets the environment, a planning module that makes decisions, and the novel coordination mechanism that links them through a task-level feedback loop. This section details the specifics of each component, as illustrated in Figure 2. The proposed framework is general and applicable to both single-agent and multi-agent settings.

### 3.1 PERCEPTION MODULE

The perception module processes raw sensory inputs and produces detection results of individual objects in the environment. Our framework supports configurations with one or more ($m \geq 1$) sensing agents. Let the perception model for each agent be a function $f$ parameterized by a set of weights $\theta_i$, where $i \in \{1, \ldots, m\}$. Given the sensory input $data_i$ for each agent, the full perception process can be formalized as the union of the outputs from all individual models:

$$\mathcal{O} = \bigcup_{i=1}^{m} f(data_i; \theta_i) \tag{1}$$

The final perception result, $\mathcal{O} = \{o_j\}_{j=1}^{n}$, is a set of $n$ detected objects. Each object $o_j$ is represented by a state vector that includes its position, dimensions, orientation, and velocity, along with its classification. Specifically, this vector is defined as $o_j = \{x, y, z, w, l, h, \sin yaw, \cos yaw, v_x, v_y, v_z, \text{class\_label}, \text{confidence\_score}\}$. This structured representation is then passed to the planning module.

## 3.2 PLANNING MODULE

The planning module is responsible for processing the perception results $\mathcal{O}$, provided by the perception module, to select an optimal action for the agent. This module is implemented as a policy, $\pi$, parameterized by a set of weights, $\phi_\pi$. A key feature of our design is that for any given state, the policy produces a dual output: the selected action to be executed in the environment, $a_t^*$, and its corresponding Predicted Reward (PR), $R_p$. The Predicted Reward represents the agent's internal belief about the immediate outcome of its action based on the perceived state. This process can be formalized as:

$$(a_t^*, R_p) = \pi(\mathcal{O}; \phi_\pi) \tag{2}$$

In multi-agent settings, the planning module employs a Multi-Agent Reinforcement Learning (MARL) algorithm, where agents learn a coordinated policy that seeks a Nash Equilibrium.

## 3.3 COORDINATED PERCEPTION AND CONTROL FRAMEWORK

Our OEL-based coordinated perception and control framework establishes a task-level feedback loop that links the outcomes of the planning module's actions back to the perception module, enabling efficient online evolution. This selective update process is a key advantage over end-to-end models, which require full backpropagation for every learning iteration. The framework, applicable to both single and multi-agent settings, operates as follows.

### 3.3.1 ACTION EXECUTION AND ENVIRONMENTAL FEEDBACK

The process begins when the action $a_t^*$, selected by the planning module, is executed in the environment. For safety and practicality, this "real environment" is a high-fidelity simulator (e.g., CARLA Dosovitskiy et al. (2017), NAVSIM Dauner et al. (2024)) that can provide ground-truth feedback. Upon execution, the environment returns an Actual Reward (AR), which we denote as $R_a$. This reward serves as the ground-truth task-level feedback for the agent's action at timestep $t$.

### 3.3.2 DISCREPANCY DETECTION

Then we compare $R_p$ and $R_a$ to decide whether to trigger update. For this comparison to be meaningful, both reward functions must be based on the same, well-defined evaluation scheme. (e.g., penalties for collisions, rewards for reaching the goal). The crucial difference is their source of information: $R_p$ is calculated based on the perceived state from the perception module, while $R_a$ is based on the ground-truth state from the simulator. Therefore, a discrepancy between them directly implies a mismatch between the agent's belief and reality, which we attribute to a perception error. An update is triggered if the absolute difference between the rewards exceeds a predefined threshold, $\tau$. We define a decision function $D_t$ at each timestep $t$:

$$D_t = \begin{cases} 1 & \text{if } |R_p - R_a| > \tau \quad \text{(Trigger Update)} \\ 0 & \text{otherwise} \quad \text{(No Update)} \end{cases} \tag{3}$$

### 3.3.3 ERROR ATTRIBUTION AND DATA RELABELING

When an update is triggered ($D_t = 1$), the system uses the agent's ego-state at that moment, $s_t$ (e.g., its position and orientation), to attribute the error. A new label, $l_\text{new}$, is generated to correct the perception error. For example, if a collision occurred in a grid cell that was perceived as empty, that cell is now given the label "obstacle". This new label is added to the set of existing labels $\mathcal{L}_t$ for the original sensory data $data_t$. We can formalize this relabeling process as a function `Relabel`:

$$(data_t, \mathcal{L}_t') = \texttt{Relabel}(data_t, \mathcal{L}_t, s_t, R_a) \tag{4}$$

where $\mathcal{L}_t' = \mathcal{L}_t \cup \{l_\text{new}\}$ is the newly corrected set of labels.

### 3.3.4 DATA BUFFERING AND ASYNCHRONOUS PERCEPTION UPDATE

The new data pair, $(data_t, \mathcal{L}_t')$, is then added to a memory buffer, $\mathcal{B}$. This process does not interrupt the agent's ongoing task execution.

$$\mathcal{B} \leftarrow \mathcal{B} \cup \{(data_t, \mathcal{L}_t')\} \tag{5}$$

Finally, the perception module, with parameters $\theta_i$, where $i \in \{1, ..., m\}$, is updated by training on mini-batches of data sampled from this buffer. This update can be performed asynchronously:

$$\theta_i \leftarrow \theta_i - \eta \nabla_{\theta_i} \mathcal{L}_{\text{perception}}(\mathcal{B}) \tag{6}$$

where $\eta$ is the learning rate and $\mathcal{L}_{\text{perception}}$ is the loss function for the perception model.

## 4 EXPERIMENTS

Our experimental evaluation is designed to answer two primary research questions. First, regarding performance and effectiveness, *how capable is our proposed coordinated perception and control framework, and how does its learning efficiency compare to other methods?* Second, concerning scalability, *how does the training cost of our paradigm scale during training as model size and environmental complexity increase, relative to baselines?* To address these questions, we designed a comprehensive suite of experiments detailed below.

### 4.1 EXPERIMENTAL SETUP

#### 4.1.1 ENVIRONMENTS

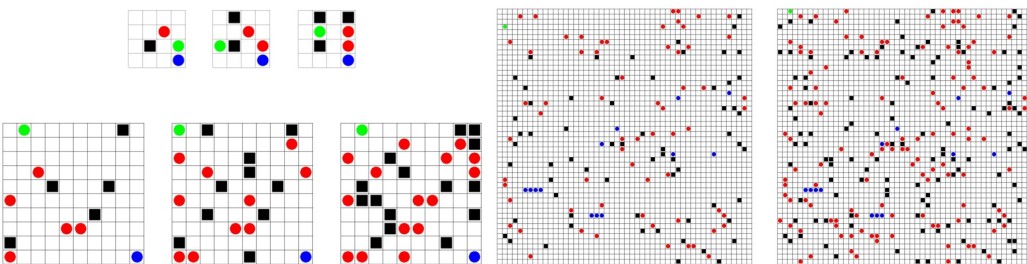

Figure 3: **The experimental grid-world environment.** The learning agent (green circle) must navigate to the goal (blue circle), avoiding both known obstacles (red circles) and unknown obstacles (black squares). The unknown obstacles are designed to be initially invisible to the perception model, creating a non-stationary challenge that explicitly tests the system's online evolutive learning capabilities.

We conduct experiments in a configurable grid-world environment (Figure 3) with varying sizes and complexities, from 4×4 to 50×50 grids. Agents, shown as green circles, aim to reach a goal (blue circle) while avoiding known obstacles (red circles) and unknown obstacles (black squares). Unknown obstacles simulate environmental non-stationarity and test the online evolutive learning of our paradigm, as agents can learn to recognize them through task-level feedback. Experiments are performed in both single-agent and multi-agent settings, with the latter requiring agents to avoid collisions with each other.

#### 4.1.2 BASELINES

We evaluate three paradigms. The end-to-end baseline is a monolithic model mapping raw inputs to actions, with parameters matched to our architecture for fairness. Our OEL paradigm features a modular architecture with a closed-loop feedback mechanism for selective perception updates. To isolate the effect of feedback, we include a decoupled ablation baseline (No OEL) with the same modular architecture but static perception after pre-training. For both OEL and No OEL, the perception module is YOLOv5 and the control module is a 3-layer MLP trained with DQN.

#### 4.1.3 EVALUATION METRICS

**1) Accuracy.** Defined as the ratio of correctly identified obstacles (known and unknown) to the total number of obstacles. An increasing trend indicates successful online evolutive learning. **2) Rewards.** Cumulative rewards measure task performance; higher rewards indicate better performance. **3) Episodes to Convergence.** Total episodes required for the policy to converge, defined

as reaching the goal in 50 consecutive episodes. Fewer episodes indicate faster learning. **4) Total FLOPs.** Total floating-point operations from training start to convergence, quantifying training cost. Detailed calculation is in the Appendix.

## 4.2 VERIFYING THE OEL PARADIGM

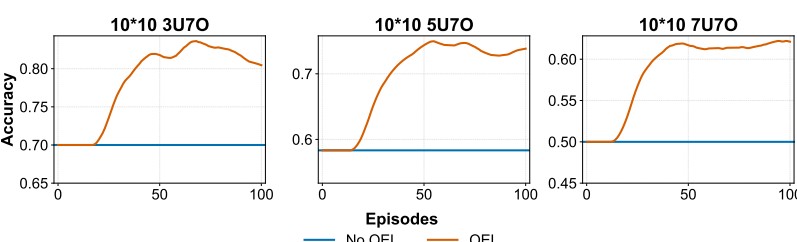

Figure 4: **Improvement of perception accuracy.** This figure compares the perception accuracy of our paradigm (OEL) against the decoupled (No OEL) baseline across scenarios with increasing numbers of unknown objects. While the baseline's accuracy remains static, our OEL approach demonstrates a clear and significant improvement over training episodes, validating the effectiveness of the task-level feedback loop. U: Unknown Objects. O: Obstacles.

To validate the OEL paradigm, we analyze its effect on the perception module by comparing our paradigm (OEL) with the decoupled baseline (No OEL), as end-to-end models do not allow independent evaluation of perception. As shown in Figure 4, No OEL accuracy remains static, while OEL accuracy steadily improves, achieving an overall 20.99% increase. This demonstrates that control-agent feedback effectively supervises and drives the online evolution of the perception.

## 4.3 ANALYSIS OF TASK PERFORMANCE AND LEARNING EFFICIENCY

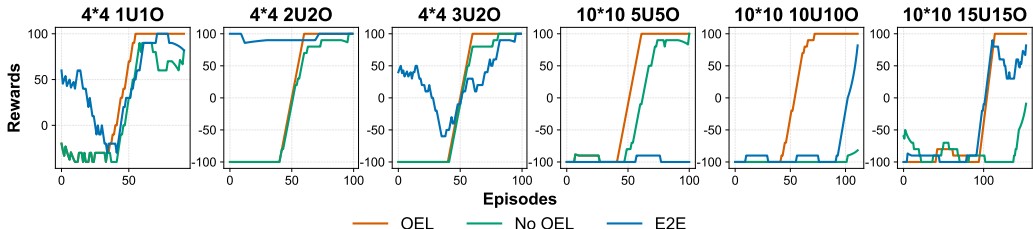

Figure 5: **Comparison of task performance and learning efficiency across paradigms.** The plots show the cumulative reward for our paradigm (OEL), the decoupled baseline (No OEL), and the end-to-end (E2E) paradigm in various scenarios. Ours consistently converge to a successful policy in significantly fewer episodes than the other two baselines, demonstrating its superior performance and efficiency. U: Unknown Objects. O: Obstacles.

Table 1: **Performance comparison of different paradigms under various gridworld scenarios.** OEL generally achieves higher average rewards and converges in fewer episodes. U: Unknown Objects. O: Obstacles.

|  | Paradigm | 4*4 1U1O | 4*4 2U2O | 4*4 3U2O | 10*10 5U5O | 10*10 10U10O | 10*10 15U15O |
|---|---|---|---|---|---|---|---|
| **Rewards ↑** | No OEL | 0.32 | 0.33 | 0.59 | 0.27 | -0.97 | -0.87 |
|  | E2E | 0.46 | **0.67** | 0.42 | -0.12 | -0.10 | -0.05 |
|  | OEL | **0.55** | 0.41 | **0.69** | **0.61** | **0.59** | **0.10** |
| **Episodes to Convergence ↓** | No OEL | 131 | 179 | 120 | 240 | 236 | 319 |
|  | E2E | **91** | 108 | 112 | 202 | 167 | 204 |
|  | OEL | 94 | **99** | **99** | **100** | **111** | **153** |

Having validated the OEL mechanism, we evaluate its impact on task performance and learning efficiency. Using cumulative reward curves and episodes to convergence, Figure 5 and Table 1

compare OEL, No OEL, and E2E baselines across scenarios. OEL achieves an average final reward $2.33\times$ higher than E2E and $10.89\times$ higher than No OEL, while converging 25.79% faster than E2E and 46.45% faster than No OEL. These results demonstrate that OEL learns more effective policies and reaches them more efficiently. Additionally, E2E outperforms No OEL ($4.67\times$ higher reward, 38.58% faster convergence), indicating that joint optimization is more effective than a static perception module, despite higher computational cost.

## 4.4 EMPIRICAL STUDY OF TRAINING COST AND SCALING LAWS

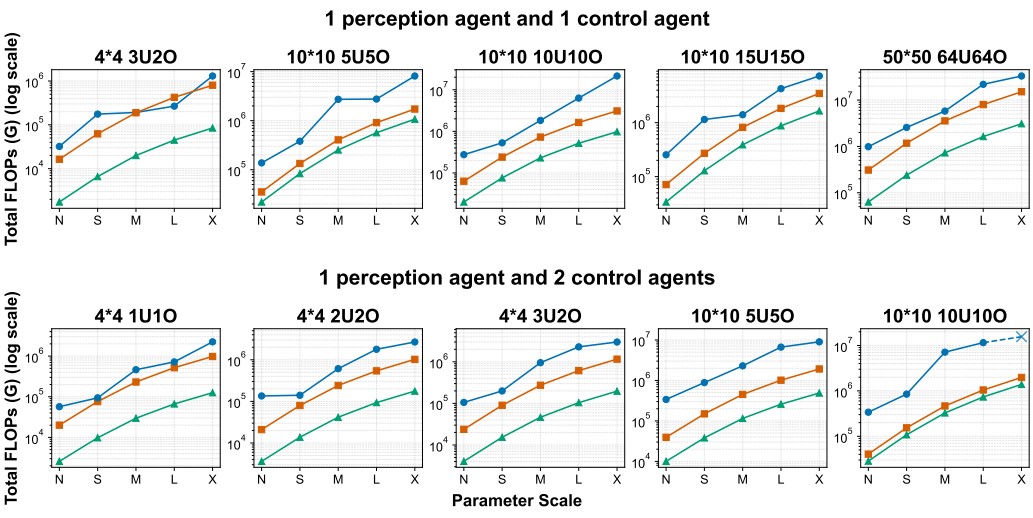

Figure 6: **Training cost vs. model scale across different scenarios and agent counts.** This figure plots the total FLOPs (log scale) as a function of model parameter scale. The results consistently demonstrate a substantial efficiency gap, with the E2E model's computational load often being half an order of magnitude greater than our OEL approach. Moreover, the dashed line indicates cases where the E2E model failed to converge. U: Unknown Objects. O: Obstacles.

Our empirical study across varying agent counts reveals clear differences in computational scaling between our paradigm, the end-to-end baseline, and the No OEL baseline. As shown in Figure 6 and Figure 7, E2E costs rise sharply with model size and environmental complexity, averaging more total FLOPs than OEL (quantitative results in Appendix Table 4 and Table 5) and often failing to converge in challenging scenarios. In contrast, OEL exhibits controlled growth, demonstrating superior efficiency and robustness. To quantify these behaviors, we fit the data to a power-law model. For model parameter scaling, the relationship is defined as:

$$C(P) = \beta \cdot P^{\alpha}, \tag{7}$$

where $C$ is the total training cost (Total FLOPs), $P$ is the number of model parameters, $\beta$ is a scaling coefficient, and $\alpha$ is the scaling exponent. The fitted scaling laws are:

$$C_{\text{E2E}}(P) \approx 1.95 \cdot P^{1.27}, \tag{8}$$

$$C_{\text{OEL}}(P) \approx 0.031 \cdot P^{1.11}, \tag{9}$$

$$C_{\text{No OEL}}(P) \approx 0.021 \cdot P^{1.01}. \tag{10}$$

These results confirm that the No OEL baseline scales almost linearly with parameters ($\alpha \approx 1.01$), OEL exhibits slightly super-linear yet controlled growth ($\alpha \approx 1.11$), and the E2E baseline grows the fastest ($\alpha \approx 1.27$), highlighting its inferior scalability.

We also analyze the effect of environmental complexity using a Environmental Complexity Index ($E$) ranging from 1 to 8, with a power-law formulation:

$$C(E) = \gamma \cdot E^{\delta}, \tag{11}$$

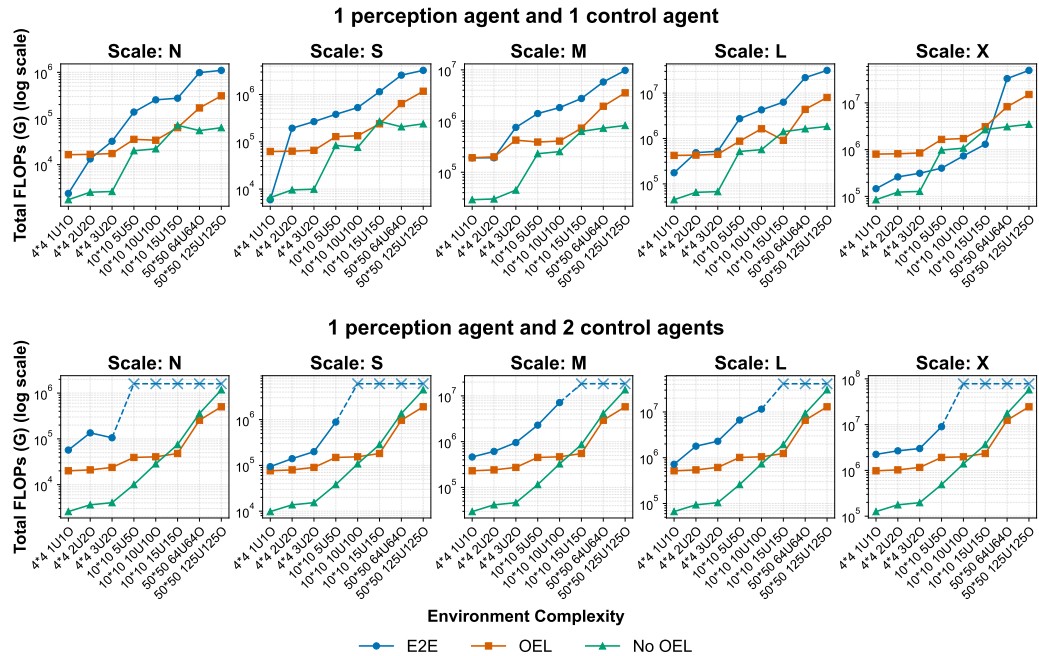

Figure 7: **Training cost vs. environmental complexity across different model scales and agent counts.** This figure plots the total FLOPs (log scale) against increasing scenario complexity. The results consistently show that as the environment becomes more challenging, the computational cost of the E2E paradigm increases dramatically, while our OEL paradigm maintains a more controlled growth. The dashed line for the E2E model indicates its failure to converge in the most complex scenarios. U: Unknown Objects. O: Obstacles.

where $\gamma$ is a scaling coefficient and $\delta$ quantifies sensitivity to environmental complexity. The fitted scaling laws are:

$$C_{\text{E2E}}(E) \approx 71805.2 \cdot E^{2.70}, \tag{12}$$

$$C_{\text{OEL}}(E) \approx 288303.7 \cdot E^{0.95}, \tag{13}$$

$$C_{\text{No OEL}}(E) \approx 37015.5 \cdot E^{2.06}. \tag{14}$$

Here, OEL demonstrates the most desirable near-linear scaling with $\delta \approx 0.95$, maintaining stability under increasing complexity. In contrast, No OEL scales quadratically ($\delta \approx 2.06$), while E2E is the most sensitive with near-cubic scaling ($\delta \approx 2.70$).

It is important to contextualize the No OEL baseline, while consuming the fewest FLOPs by omitting adaptive updates, suffers from static perception, poor adaptation to unknown obstacles, and slower convergence, limiting robustness and task performance (Sections 4.2 and 4.3). Thus, although computationally cheapest, it is unsuitable for dynamic environments. The modest extra cost of OEL provides essential adaptability and scalable efficiency.

## 5 CONCLUSION

In this work, we address the computational scaling issues of end-to-end paradigm. We introduce the coordinated perception and control framework, where a task-level feedback loop enables selective perception updates, retaining the benefits of end-to-end learning while improving efficiency. Extensive experiments show that our approach requires on far less training cost than end-to-end baselines, with nearly linear or sub-linear scaling ($\alpha \approx 1.11$, $\delta \approx 0.95$) compared to the super-linear growth of end-to-end models ($\alpha \approx 1.27$, $\delta \approx 2.70$). These results hold across model scales and environmental complexities, providing a concrete design principle for scalable intelligent systems. Future work should validate these efficiency gains in high-fidelity simulators and real-world robotic platforms.

## 6 ETHICS STATEMENT

We have read and adhered to the ICLR Code of Ethics. This research is foundational, focusing on the computational efficiency and scaling properties of different learning paradigms for autonomous agents. All experiments were conducted in a self-contained, synthetic grid-world environment, and as such, did not involve human subjects, real-world data collection, or personally identifiable information, thus mitigating concerns related to privacy and the need for Institutional Review Board (IRB) approval. The primary goal of our work is to contribute to the development of more computationally sustainable and efficient AI, which we consider a socially responsible objective. The abstract nature of our task environment means that societal biases are not a direct factor in this study.

## 7 REPRODUCIBILITY STATEMENT

We are committed to ensuring the reproducibility of our research. The main paper's Methods section (Section 3) details our proposed architecture and core mechanisms. The Appendix further provides all necessary implementation details for replication, including network architectures and hyperparameters (Appendix A.1.1 and A.1.2), the pseudocode for our algorithm (Appendix A.2), and the formalism for our training cost metric (Appendix A.1.3). To this end, the full source code for the experimental environment, model implementations, and plotting code and scripts are provided in the supplementary materials to replicate the empirical results.

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

# A APPENDIX

## A.1 IMPLEMENTATION DETAILS

### A.1.1 MODEL SCALE

To systematically analyze the scaling properties of the two learning paradigms, we instantiated five different versions of each architecture, corresponding to increasing scales of complexity, denoted as N, S, M, L, and X. These scales are based on the five standard variants of the YOLOv5 architecture, from YOLOv5n (nano) to YOLOv5x (extra-large), which serve as the perception backbone in our experiments. The specific parameters are detailed in Table 2. As shown in Table 2, the total parameter count for the end-to-end models ranges from 1.39M (N scale) to 55.8M (X scale). Our coordinated perception and control models are composed of a full YOLOv5 perception module of the corresponding scale and a separate, fixed-size planning module (a 3-layer MLP). As detailed in Table 2, the total parameter count for our framework and end-to-end models are comparable, with our architecture having 1.28 to 1.73 more parameters than the end-to-end baselines across the different scales. This comparability in model capacity ensures that our study provides a fair evaluation of the two paradigms, allowing us to directly assess the architectural impact on the total training cost.

Table 2: Parameter size for different model scales.

| Scale | Total Parameters | |
|:---:|:---:|:---:|
| | **End-to-End** | **Coordinated Perception and Control** |
| **N** | 1,386,820 | 1,769,329 |
| **S** | 4,065,956 | 7,030,417 |
| **M** | 14,019,908 | 20,883,441 |
| **L** | 31,925,060 | 46,154,449 |
| **X** | 55,796,836 | 86,238,001 |

### A.1.2 TRAINING HYPERPARAMETERS

All models were trained using the Adam optimizer. The key hyperparameters used across our experiments are detailed in Table 3.

Table 3: Key hyperparameters used for training the different paradigms.

| Hyperparameter | Value |
|:---|:---|
| *General RL Parameters* | |
| Optimizer | Adam |
| Learning Rate (LR) | 0.008 |
| Discount Factor ($\gamma$) | 0.999 |
| Batch Size | 32 |
| Replay Buffer Size | 100,000 |
| Target Network Update Freq. | 10 episodes |
| *$\epsilon$-greedy Strategy* | |
| Initial Exploration Episodes | 100 |
| Epsilon Decay Rate | 0.99 per episode |
| Epsilon Coefficient | 0.4 |
| Minimum Epsilon | 0.0001 |
| *OEL-Specific Parameter* | |
| Update Threshold ($\tau$) | 0.4 |

### A.1.3 Total FLOPs Calculation Formula

This section provides a detailed breakdown of the methodology used to calculate the Total FLOPs for the end-to-end and our paradigms. This metric quantifies the total computational resources consumed from the beginning of a training run until the agent's policy converges.

**End-to-End Paradigm.** The total training cost for the end-to-end paradigm, $C_{\text{E2E}}$, is composed of the cost of continuous agent-environment interaction and the cost of model optimization via backpropagation. At every timestep, the agent performs one forward pass through the entire network to select an action. Let $S$ be the total number of timesteps until convergence and $F_{\text{E2E}}$ be the FLOPs for a single forward pass. The total interaction cost is:

$$C_{\text{interact}} = S \times F_{\text{E2E}} \tag{15}$$

We assume the model begins learning after an initial exploration phase of $\tau$ steps. For every step thereafter ($S - \tau$ steps), a learning update is performed. A single update involves a forward and backward pass for a batch of experiences. Approximating the backward pass as twice the cost of a forward pass, the cost of one optimization step is $3 \times F_{\text{E2E}} \times B_{\text{E2E}}$, where $B_{\text{E2E}}$ is the batch size. The total optimization cost is:

$$C_{\text{optim}} = (S - \tau) \times (3 \times F_{\text{E2E}} \times B_{\text{E2E}}) \tag{16}$$

The total FLOPs for the E2E paradigm is the sum of these two components:

$$C_{\text{E2E}} = S \times F_{\text{E2E}} + (S - \tau) \times 3 \times F_{\text{E2E}} \times B_{\text{E2E}} \tag{17}$$

**Online Evolutive Learning Paradigm.** The total computational cost for our OEL paradigm, $C_{\text{OEL}}$, is the sum of three distinct components: the cost of the main agent interaction loop, the cost of optimizing the planning module, and the cost of selectively retraining the perception module. At every timestep, the agent performs a forward pass through both the perception module ($F_{\text{perc}}$) and the planning module ($F_{\text{plan}}$). For a total of $S$ timesteps, the interaction cost is:

$$C_{\text{interact}} = S \times (F_{\text{perc}} + F_{\text{plan}}) \tag{18}$$

Similar to the end-to-end paradigm, the planning module begins learning after $\tau$ steps. The cost for each update is $3 \times F_{\text{plan}} \times B_{\text{plan}}$, where $B_{\text{plan}}$ is the batch size for the planning module's replay buffer. The total cost for optimizing the planner is:

$$C_{\text{optim\_plan}} = (S - \tau) \times (3 \times F_{\text{plan}} \times B_{\text{plan}}) \tag{19}$$

This cost is incurred when the perception module is updated using the data collected in the perception buffer. This is a separate process. Let's assume it occurs once during the training run. The total cost is determined by the batch size ($B_{\text{perc}}$), the number of training steps per epoch ($S_{\text{epoch}}$), and the total number of epochs ($N_{\text{epochs}}$).

$$C_{\text{retrain\_perc}} = 3 \times F_{\text{perc}} \times B_{\text{perc}} \times S_{\text{epoch}} \times N_{\text{epochs}} \tag{20}$$

The total FLOPs for our OEL paradigm is the sum of these three components:

$$C_{\text{OEL}} = C_{\text{interact}} + C_{\text{optim\_plan}} + C_{\text{retrain\_perc}} \tag{21}$$

### A.1.4 Architectural Design for Multi-Agent End-to-End Paradigm

To ensure a fair and strong comparison, we designed a multi-agent end-to-end baseline based on the common paradigm of a shared network body with multiple output heads.

**Network Architecture.** The end-to-end model takes a single RGB image of the entire grid-world as input. This input is first processed by a shared feature extraction backbone, which is identical to the YOLOv5 backbone used in the corresponding OEL experiment. The resulting feature map is then flattened into a vector. This shared feature vector is subsequently fed into multiple parallel decision heads. Each head is a dedicated MLP responsible for a single agent, and it outputs the Q-values for that agent's possible actions.

**Training Algorithm.** The model is trained using the DQN algorithm with a centralized training approach. At each timestep, a single transition containing the overall state, the joint action of all agents, and the individual rewards for each agent is stored in a unified replay buffer. During the learning step, a batch of these bundled transitions is sampled. A composite loss is calculated (e.g., the sum of the individual agents' Q-losses), and this single loss signal is backpropagated through the entire network. This means the gradients flow from all individual heads back through the shared feature extractor.

## A.2 ALGORITHM PSEUDOCODE

Algorithm 1 presents the detailed pseudocode for the training process of our coordinated perception and control framework, which is enabled by OEL. The algorithm formalizes the interplay between two distinct learning loops that operate concurrently. This mechanisms, particularly the conditional and asynchronous nature of the perception update, is the key to the framework's computational efficiency and favorable scaling properties as demonstrated in the main paper.

---

**Algorithm 1:** Coordinated Perception and Control with OEL

---

**Input:** Environment $env$, update threshold $\tau$, episodes $N_{\text{episodes}}$, steps $T_{\text{steps}}$
**Initialize**
    Perception model $f$ with parameters $\theta_i$
    Planning policy $\pi$ with parameters $\phi_\pi$
    Perception update buffer $\mathcal{B}_p \leftarrow \emptyset$
    RL replay buffer $\mathcal{B}_{rl} \leftarrow \emptyset$
**for** *episode = 1 to $N_{episodes}$* **do**
    Reset environment and get initial sensory data $data_0$
    **for** *t = 0 to $T_{steps} - 1$* **do**
        // 1. Perception
        Get perceived state representation $\mathcal{O}_t \leftarrow \bigcup_{i=1}^m f(data_i^t; \theta_i)$
        // 2. Planning
        Select action and get Predicted Reward $(a_t^*, R_p) \leftarrow \pi(\mathcal{O}_t; \phi_\pi)$
        // 3. Execution and Environmental Feedback
        Execute action $a_t^*$ in $env$ to get next data $data^{t+1}$, Actual Reward $R_a$, and done signal
        // 4. Standard RL Update for Planning Module
        Get next perceived state $\mathcal{O}_{t+1} \leftarrow \bigcup_{i=1}^m f(data_i^{t+1}; \theta_i)$
        Store transition $(\mathcal{O}_t, a_t^*, R_a, \mathcal{O}_{t+1})$ in RL replay buffer $\mathcal{B}_{rl}$
        Update planning policy parameters $\phi_\pi$ using a mini-batch from $\mathcal{B}_{rl}$
        // 5. Coordinated Feedback Loop for Perception Module
        **if** $|R_p - R_a| > \tau$ **then**
            // Discrepancy detected, trigger perception update
            Get agent's current ego-state $s_t$ from $env$
            Generate corrected labels $(data_t, \mathcal{L}_t') \leftarrow \text{Relabel}(data_t, \mathcal{O}_t, s_t, R_a)$
            Store the corrected data pair in the perception update buffer
            $\mathcal{B}_p \leftarrow \mathcal{B}_p \cup \{(data_t, \mathcal{L}_t')\}$
        **end**
        // 6. Perception Update
        **if** *it is time to update perception model* **then**
            Update perception model parameters $\theta_i$ using a mini-batch from $\mathcal{B}_p$
        **end**
        $data_t \leftarrow data_{t+1}$
        **if** *done* **then**
            break
        **end**
    **end**
**end**

---

## A.3 EXTENDED EXPERIMENTAL RESULTS

### A.3.1 SCALABILITY WITH RESPECT TO AGENT COUNTS

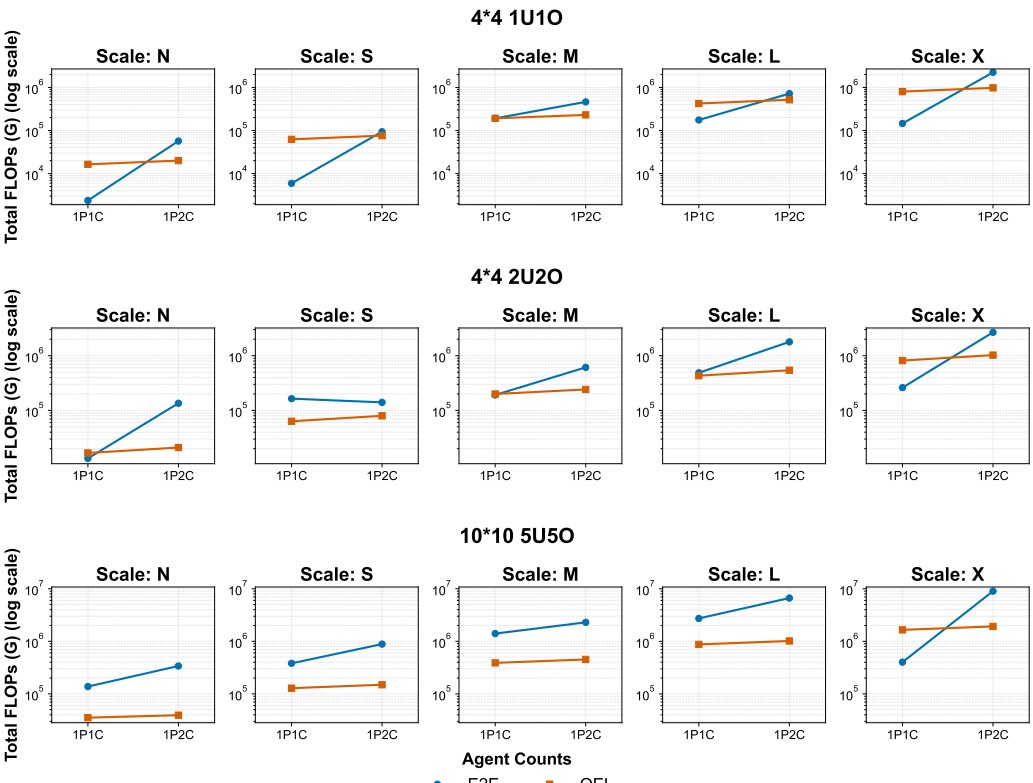

Figure 8: **Training cost scaling with increasing agent counts.** This figure compares the Total FLOPs for the OEL and E2E paradigms when scaling from a single-agent (1P1C) to a multi-agent (1P2C) setting. The results show that our paradigm's computational cost increases only marginally with an additional agent, whereas the E2E baseline's cost grows substantially. This highlights the superior scalability and robustness of our paradigm for multi-agent systems. U: Unknown Objects. O: Obstacles. P: Perception Agent. C: Control Agent.

To further investigate the scalability of our framework, we directly analyze how the computational cost of each paradigm scales with an increase in the number of control agents. As shown in Figure 8, we compare the Total FLOPs required for convergence when transitioning from a single-agent (1-perception-1-control) to a multi-agent (1-perception-2-control) setup across several scenarios. The results reveal a critical architectural advantage of our approach. The total computational cost for our paradigm shows only a marginal increase with the addition of a second agent. However, the end-to-end baseline's computational requirements increased substantially under the same conditions. This significant growth occurs because the end-to-end model must learn a much more complex, high-dimensional joint policy for multiple agents, which dramatically increases the optimization difficulty and the total computation needed to converge. This direct comparison underscores the superior scalability of our approach, positioning it as a more viable and efficient architecture for developing multi-agent systems.

### A.3.2 FULL QUANTITATIVE RESULTS

We provide the complete and detailed quantitative results in Table 4 and Table 5. Table 4 and Table 5 tabulate the total computational cost, measured in Total FLOPs (G), required for each of the three paradigms (E2E, OEL, and No OEL) to achieve a convergent policy.

Table 4: Total FLOPs (G) comparison under different parameter scales and scene complexities with 1 perception agent and 1 control agent.

| Scale | Scenairo Complexity | Total FLOPs (G) | | |
|---|---|---|---|---|
| | | E2E | OEL | No OEL |
| N | 4×4, 1 unknown, 1 obstacle | 2374.78 | 16414.86 | 1734.70 |
| | 4×4, 2 unknown, 2 obstacles | 13276.63 | 16674.02 | 2524.72 |
| | 4×4, 3 unknown, 2 obstacles | 32080.79 | 17365.81 | 2612.50 |
| | 10×10, 5 unknown, 5 obstacles | 137836.44 | 35312.64 | 20018.02 |
| | 10×10, 10 unknown, 10 obstacles | 253591.14 | 33805.75 | 21982.62 |
| | 10×10, 15 unknown, 15 obstacles | 274784.26 | 63264.3 | 71220.93 |
| | 50×50, 64 unknown, 64 obstacles | 985637.56 | 169252.38 | 54546.91 |
| | 50×50, 125 unknown, 125 obstacles | 1102837.91 | 309217.59 | 63128.45 |
| S | 4×4, 1 unknown, 1 obstacle | 5942.70 | 62350.76 | 6590.20 |
| | 4×4, 2 unknown, 2 obstacles | 194030.22 | 63345.32 | 9591.52 |
| | 4×4, 3 unknown, 2 obstacles | 267072.18 | 65973.46 | 9925.00 |
| | 10×10, 5 unknown, 5 obstacles | 381012.98 | 128429.50 | 83512.92 |
| | 10×10, 10 unknown, 10 obstacles | 529006.8 | 134154.24 | 76049.32 |
| | 10×10, 15 unknown, 15 obstacles | 1145056.76 | 240343.8 | 270571.38 |
| | 50×50, 64 unknown, 64 obstacles | 2567632.11 | 642997.08 | 207226.06 |
| | 50×50, 125 unknown, 125 obstacles | 3243665.35 | 1174730.94 | 239827.7 |
| M | 4×4, 1 unknown, 1 obstacle | 191556.00 | 191791.12 | 29040.32 |
| | 4×4, 2 unknown, 2 obstacles | 191556.00 | 199748.36 | 30050.00 |
| | 4×4, 3 unknown, 2 obstacles | 752051.36 | 424037.46 | 44811.70 |
| | 10×10, 5 unknown, 5 obstacles | 1402377.72 | 388847.00 | 230255.12 |
| | 10×10, 10 unknown, 10 obstacles | 1830762.04 | 406179.84 | 252852.72 |
| | 10×10, 15 unknown, 15 obstacles | 2758481.52 | 727690.8 | 627419.96 |
| | 50×50, 64 unknown, 64 obstacles | 5792745.66 | 1946807.28 | 726128.2 |
| | 50×50, 125 unknown, 125 obstacles | 9773611.08 | 3556742.04 | 819211.08 |
| L | 4×4, 1 unknown, 1 obstacle | 175742.58 | 424037.46 | 44811.70 |
| | 4×4, 2 unknown, 2 obstacles | 486965.34 | 430732.22 | 65219.92 |
| | 4×4, 3 unknown, 2 obstacles | 522586.38 | 448602.91 | 67487.50 |
| | 10×10, 5 unknown, 5 obstacles | 2725540.80 | 873288.25 | 517116.22 |
| | 10×10, 10 unknown, 10 obstacles | 4267269.18 | 1634277.3 | 567866.82 |
| | 10×10, 15 unknown, 15 obstacles | 6303889.56 | 912215.04 | 1409085.01 |
| | 50×50, 64 unknown, 64 obstacles | 21863799.49 | 4372218.18 | 1630767.95 |
| | 50×50, 125 unknown, 125 obstacles | 31567783.98 | 7987874.49 | 1839817.23 |
| X | 4×4, 1 unknown, 1 obstacle | 145841.92 | 802050.48 | 84759.60 |
| | 4×4, 2 unknown, 2 obstacles | 260825.60 | 814713.36 | 123360.96 |
| | 4×4, 3 unknown, 2 obstacles | 312360.96 | 848515.08 | 127650.00 |
| | 10×10, 5 unknown, 5 obstacles | 401288.32 | 1651791.00 | 978105.36 |
| | 10×10, 10 unknown, 10 obstacles | 733014.4 | 1725419.52 | 1074098.16 |
| | 10×10, 15 unknown, 15 obstacles | 1302055.04 | 3091172.4 | 2665229.88 |
| | 50×50, 64 unknown, 64 obstacles | 33076124 | 8269881.84 | 3084534.6 |
| | 50×50, 125 unknown, 125 obstacles | 49578649.12 | 15108756.12 | 3479943.24 |

Table 5: Total FLOPs (G) comparison under different parameter scales and scene complexities with 1 perception agent and 2 control agents.

| Scale | Scenario Complexity | Total FLOPs (G) | | |
|---|---|---|---|---|
| | | E2E | OEL | No OEL |
| N | 4×4, 1 unknown, 1 obstacle | 56930.53 | 20051.46 | 2576.97 |
| | 4×4, 2 unknown, 2 obstacles | 135122.01 | 20983.6 | 3617.79 |
| | 4×4, 3 unknown, 2 obstacles | 105676.47 | 23771.66 | 4033.7 |
| | 10×10, 5 unknown, 5 obstacles | - | 39283.64 | 10019.46 |
| | 10×10, 10 unknown, 10 obstacles | - | 40472.85 | 28419.82 |
| | 10×10, 15 unknown, 15 obstacles | - | 47871.45 | 75039.36 |
| | 50×50, 64 unknown, 64 obstacles | - | 255249.61 | 357939.67 |
| | 50×50, 125 unknown, 125 obstacles | - | 501409.81 | 1177863.39 |
| S | 4×4, 1 unknown, 1 obstacle | 93947.51 | 76176.36 | 9790.02 |
| | 4×4, 2 unknown, 2 obstacles | 140331.21 | 79717.6 | 13744.14 |
| | 4×4, 3 unknown, 2 obstacles | 199297.09 | 90309.56 | 15324.2 |
| | 10×10, 5 unknown, 5 obstacles | 884830.63 | 149240.24 | 38064.36 |
| | 10×10, 10 unknown, 10 obstacles | - | 153758.1 | 107968.12 |
| | 10×10, 15 unknown, 15 obstacles | - | 181865.7 | 285077.76 |
| | 50×50, 64 unknown, 64 obstacles | - | 969704.26 | 1359828.22 |
| | 50×50, 125 unknown, 125 obstacles | - | 1904877.46 | 4474753.74 |
| M | 4×4, 1 unknown, 1 obstacle | 462250 | 230639.76 | 29641.32 |
| | 4×4, 2 unknown, 2 obstacles | 611500 | 241361.6 | 41613.24 |
| | 4×4, 3 unknown, 2 obstacles | 954575 | 273430.96 | 46397.2 |
| | 10×10, 5 unknown, 5 obstacles | 2293875 | 451855.84 | 115247.76 |
| | 10×10, 10 unknown, 10 obstacles | 7128000 | 465534.6 | 326895.92 |
| | 10×10, 15 unknown, 15 obstacles | - | 550636.2 | 863132.16 |
| | 50×50, 64 unknown, 64 obstacles | - | 2935981.16 | 4117162.52 |
| | 50×50, 125 unknown, 125 obstacles | - | 5767412.36 | 13548246.84 |
| L | 4×4, 1 unknown, 1 obstacle | 719097.6 | 517980.06 | 66569.67 |
| | 4×4, 2 unknown, 2 obstacles | 1788157.15 | 542059.6 | 93456.69 |
| | 4×4, 3 unknown, 2 obstacles | 2291620.5 | 614082.26 | 104200.7 |
| | 10×10, 5 unknown, 5 obstacles | 6653777 | 1014796.04 | 258828.06 |
| | 10×10, 10 unknown, 10 obstacles | 11545530.1 | 1045516.35 | 734156.02 |
| | 10×10, 15 unknown, 15 obstacles | - | 1236640.95 | 1938456.96 |
| | 50×50, 64 unknown, 64 obstacles | - | 6593744.71 | 9246489.37 |
| | 50×50, 125 unknown, 125 obstacles | - | 12952686.91 | 30427198.29 |
| X | 4×4, 1 unknown, 1 obstacle | 2241233.16 | 979739.28 | 125913.96 |
| | 4×4, 2 unknown, 2 obstacles | 2675803.44 | 1025284.8 | 176769.72 |
| | 4×4, 3 unknown, 2 obstacles | 3016354.55 | 1161512.88 | 197091.6 |
| | 10×10, 5 unknown, 5 obstacles | 9195332.09 | 1919447.52 | 489563.28 |
| | 10×10, 10 unknown, 10 obstacles | - | 1977553.8 | 1388627.76 |
| | 10×10, 15 unknown, 15 obstacles | - | 2339058.6 | 3666516.48 |
| | 50×50, 64 unknown, 64 obstacles | - | 12471813.48 | 17489377.56 |
| | 50×50, 125 unknown, 125 obstacles | - | 24499507.08 | 57551870.52 |

## A.4 THE USE OF LARGE LANGUAGE MODELS (LLMs)

We utilized Large Language Models (LLMs), including Google Gemini and ChatGPT, to aid and polish the writing of this manuscript. Their role was assistive and confined to language refinement, such as translating initial drafts and improving the text's clarity and grammar. All core scientific contributions, including research ideation, methodology, and analysis, were conducted entirely by the human authors, who take full responsibility for the final content.

