# OpenReview forum: "Beyond End-to-End Models: Characterizing the Favorable Scaling of Coordinated Perception and Control"
_ICLR.cc/2026/Conference — ICLR 2026 Conference Withdrawn Submission_

### Official Review · Reviewer_9ac5 · 2025-10-28

**Soundness:** 2
**Presentation:** 2
**Contribution:** 2
**Rating:** 2
**Confidence:** 5

**Summary:**

This paper introduces a coordinated perception and control framework based on Online Evolutive Learning (OEL) to address the unfavorable computational scaling of end-to-end (E2E) models as model size and environmental complexity grow—a critical challenge that is not well-addressed in current embodied AI systems. The proposed OEL framework decouples perception and control into modular components but connects them via a task-level feedback loop, where discrepancies between predicted and actual rewards trigger selective, lightweight updates to the perception module. The experiment results reveal that OEL achieves near-linear or sub-linear scaling in training cost, outperforming E2E models, while matching or exceeding their task performance.

**Strengths:**

- The paper is clearly written and well-structured.
- It introduces the OEL framework, a decoupled perception-control framework that improves training efficiency.
- The empirical results demonstrate the effectiveness of OEL.

**Weaknesses:**

- I believe the authors overclaim their contribution regarding the scaling law analysis. In lines 157-160, they claim that "previous works primarily examine model size or dataset scale rather than training cost." However, several existing studies, such as [1, 2], have already explored the relationship between task performance and training cost (i.e., FLOPs). Moreover, when model architecture, model size, and dataset size are fixed, the training cost becomes a directly computable quantity.
- OEL relies heavily on manually designed priors, limiting its generalizability. For example, the perceptual module’s output depends on predefined rules, restricting its applicability to more complex tasks like visual-based manipulation tasks.
- OEL may perform poorly in sparse reward tasks. In such tasks, using the difference between predicted and actual reward as a trigger for perception updates is problematic: since actual rewards are often identical (e.g., zero), the signal is too sparse to drive learning, and the reward predictor may collapse to always output the default value, masking perception errors and preventing correction.
- The comparison with baselines is unfair, weakening the objectivity of the conclusions. The claim that the decoupled OLE framework has better scalability than e2e methods is questionable, as OLE includes human priors, while e2e methods are purely data-driven, making it unclear whether improvements come from the framework or the priors.
- The experiments on maze environments are too simple. Given the claim of "practical guidance for building next-generation embodied AI systems that are both adaptive and computationally efficient" in lines 29-31, more experiments on complex tasks like robotic manipulation and locomotion are necessary.

[1] Scaling Laws for Neural Language Models.

[2] Scaling Laws for Pre-training Agents and World Models.

**Questions:**

- How can we use OEL to generalize beyond maze tasks, without its reliance on human-designed priors?
- Can the authors clarify whether the scalability improvements of the decoupled OLE framework are due to its design or the use of human priors, compared to purely data-driven e2e methods?
- How does OEL perform in sparse reward tasks like AntMaze or Minecraft?
- How does OEL perform in more complex real-world tasks like manipulation or locomotion?

---

### Official Review · Reviewer_wxye · 2025-11-01

**Soundness:** 2
**Presentation:** 3
**Contribution:** 2
**Rating:** 4
**Confidence:** 2

**Summary:**

The paper studies the computational scaling behavior of embodied-learning systems and advances a coordinated perception–control framework inspired by Online Evolutive Learning (OEL). The approach decouples perception and planning, then links them with a closed-loop task-level feedback signal. Figures 1–2 illustrate the contrast with standard end-to-end (E2E) pipelines and the proposed feedback loop, respectively. The empirical study is performed in a configurable grid-world across parameter scales, environment complexities. The central claim is that the proposed OEL coordination yields markedly better computational scaling than E2E.

**Strengths:**

- The paper explicitly targets training cost scaling, a dimension often overlooked in embodied AI work that focuses mainly on asymptotic task performance.
- The paper reports near-linear complexity scaling for OEL versus near-cubic for E2E, and reduced parameter‑sensitivity.

**Weaknesses:**

- The method treats large |PR−AR| as evidence of misperception (Sec. 3.3.2), yet PR is produced by the planner. With a DQN‑style controller (Sec. 4.1.2), PR is close to a value‑like quantity: it inevitably reflects planning inaccuracies and exploration noise. The paper needs an ablation that disambiguates perception vs. planning error contributions to |PR−AR|. As written, the trigger will sometimes relabel perception for what is fundamentally a control‑policy misprediction.

- Evaluation domain is abstract; generality is asserted but not demonstrated. All experiments are in a 2D grid‑world (Fig. 3). The Introduction and Sec. 3.3.1 gesture to CARLA/NAVSIM (as “real environments” for safe feedback), but the study never leaves grid‑world. Given the strong claims about favorable scaling “for next‑generation embodied AI systems,” at least one high‑fidelity sim (e.g., CARLA closed loop) is needed to support external validity.

- The paper reports single numbers/curves with no confidence intervals or seed sweeps (Fig. 5; Figs. 6–8; Tables 4–5). Because convergence and FLOPs can vary substantially across seeds in RL, the aforementioned should be provided.

- There seems to be an ambiguity around what the policy predicts. The planner outputs an action and a “Predicted Reward (PR)” representing the anticipated immediate reward (Sec. 3.2). But the controller is trained with DQN (Sec. 4.1.2), whose natural output is an action-value approximating discounted return. If PR is only the immediate reward, it is a lossy signal for discrepancies that arise from perception defects with delayed consequences; if it is actually a Q‑value, the paper’s terminology and thresholding rationale need to be clarified.

**Questions:**

see weaknesses section

---

### Official Review · Reviewer_DhAo · 2025-11-03

**Soundness:** 4
**Presentation:** 4
**Contribution:** 4
**Rating:** 6
**Confidence:** 4

**Summary:**

The paper studies the computational scaling of “coordinated perception and control” versus end-to-end (E2E) training for embodied agents. The coordinated approach decouples perception and control and uses a task-level feedback loop: the planner outputs both an action and a predicted reward; after execution, a discrepancy with the actual reward triggers selective updates of the perception module using relabeled data. YOLOv5 is used as the perception module and a 3-level MLP is used as the control module trained with DQN. Experiments in configurable grid-worlds with sizes ranging from 4x4 to 50x50, report more favorable training-cost scaling exponents for the coordinated paradigm with respect to both parameter count and environment complexity, plus faster convergence at similar or better rewards than E2E.

**Strengths:**

1. The paper is excellently written. The authors have done a great job taking a simple idea and presenting it with a clear motivation and experiment design.
2. The idea is simple to understand, use and implement. “Coordinated perception and control” is a fairly novel idea to overcome the scaling issues in E2E training paradigm.
3. Empirical efficiency: Well designed experiments and scaling law study. Reported exponents favor the coordinated approach
4. I believe, if the presented work on coordinated perception and control truly scales to realistic tasks with more complex state and action spaces, it has far reaching implications for robotics, self-driving, and potentially RL based finetuning for reasoning models.
5. Reproducible work (shared source code & implementation). Also promptly report the use of LLMs for improving the writing.

**Weaknesses:**

1. Attribution of errors to perception is somewhat unjustified. The method assumes |PR - AR| > $\tau$ indicates a perception error, then updates perception. Value prediction can be poorly calibrated even with perfect perception; mis-estimation or exploration noise can also cause large PR - AR gaps. Can the authors clarify the choice of error attribution purely to perception and provide diagnostics separating planner-value error from perception error?
2. Hyperparameter sensitivity: The hyperparam $\tau$ might be a bit problematic in more complex scenarios. It seems to be fixed at 0.4 in the experiments. How did the authors choose this and how sensitive are their experiments to $\tau$? Can this be shown empirically? Can the authors provide a consistent protocol for choosing the optimal $\tau$ to observe the reported scaling law rules?
3. Experimental validity: All results are in a stylized grid-world designed for the specific evaluation while the paper motivates autonomous driving and robotics. Claims about “fundamentally more scalable design” for embodied AI are premature without CARLA/Habitat style navigation test. I am curious why the authors did not attempt to evaluate their results on more realistic scenarios. Can you replicate the scaling study in CARLA or Habitat with sparse relabeling to show external validity?
4. Also, why is a heavy vision backbone like YOLOv5 used on a simple gridworld environment?
5. How often is the perception module in the OEL method actually retrained/updated during a run? Can these results be presented and added to the tables?

Minor:

- Figure 1 correct “Foward Pass” → Forward Pass

**Questions:**

Asked above.

Overall, despite the minor questions & issues, I think this paper is an accept for its clear motivation, simple idea that empirically works and has good potential to drive the field forward in the right direction. I am willing to increase my scores further if my questions are answered appropriately with empirical justification during the discussion phase.

---

### Official Review · Reviewer_VfJJ · 2025-11-09

**Soundness:** 2
**Presentation:** 3
**Contribution:** 2
**Rating:** 2
**Confidence:** 4

**Summary:**

This paper addresses the computational scaling challenges of E2E learning paradigms in embodied AI. The authors propose an alternative, "coordinated perception and control" architecture, based on online evolutive learning. This framework decouples the perception and control modules. Its core contribution lies in a task-level feedback loop: the planning module's predicted reward is compared against the actual reward from the environment. A significant discrepancy triggers a selective, asynchronous update of only the perception module, driven by relabeling the problematic sensory data. The authors conduct an empirical study to characterize the scaling laws of this OEL paradigm against an E2E baseline and a fully decoupled (static perception) baseline. The primary claim, based on experiments, is that OEL achieves nearly linear or sub-linear computational scaling with model size and environmental complexity, contrasting the E2E model's super-linear or near-cubic growth.

**Strengths:**

The paper targets a critical and widely recognized bottleneck in AI: the unsustainable computational and sample-efficiency costs of scaling E2E models, particularly for real-world embodied agents. A solution that offers better scaling properties would be a significant contribution.

The proposed OEL feedback loop is well-explained. The core idea of using task-level reward discrepancy as a signal to trigger selective and asynchronous perception updates is logical. It presents a clear alternative to the brute-force, full-system backpropagation required by monolithic E2E models in every iteration.

**Weaknesses:**

This paper's core architectural idea is presented as novel, but it appears to be a direct application of previously published work. The authors explicitly state their framework is "inspired by the idea of Online Evolutive Learning" and "driving agents based on the OEL paradigm in Qian et al. (2024)". The contribution thus shrinks to (a) applying this existing OEL idea to a new domain and (b) performing a scaling analysis. Given that the chosen domain is a grid-world, this contribution is limited in this way.

The authors cannot claim to provide "practical guidance for building next-generation embodied AI systems" or "a concrete design principle for scalable intelligent systems"  based on grid-world results. The observed scaling laws are highly likely to be artifacts of this simplistic environment and may not hold in any real-world setting. The authors even relegate validation in high-fidelity simulators to "future work", which is precisely what should have been the bare minimum for this submission.

Regarding the experiments, the results show this static model scales quadratically with environment complexity, which is significantly worse than the adaptive OEL model's near-linear scaling. This is deeply counter-intuitive. A static model that performs no perception updates should have a computational cost that is less sensitive to complexity than an adaptive one. The paper's explanation—that it suffers from "slower convergence" —simply highlights that the "Total FLOPs" metric is confounded by performance. This suggests the experimental setup or the metric itself is flawed, casting doubt on all the scaling exponents reported

Besides, the work is motivated by and makes broad claims about "embodied AI," "robotics," and "autonomous driving". However, the entire empirical validation is conducted in a "configurable grid-world environment". This is a discrete, low-dimensional, fully-observable toy problem that bears no meaningful resemblance to the high-dimensional, continuous, and partially-observable challenges of real-world perception and control.

**Questions:**

Please refer to the weakness part above.

---

### Note · Authors · 2025-11-12

I have read and agree with the venue's withdrawal policy on behalf of myself and my co-authors.